# Prototyping Mid-Air Display for Anywhere Robot Communication With Projected Spatial AR

### Uthman Tijani
RARE Lab
Department of Computer Science and Engineering
University of South Florida
Tampa, Florida, USA
uthmantijani@usf.edu

### Zhao Han
RARE Lab
Department of Computer Science and Engineering
University of South Florida
Tampa, Florida, USA
zhaohan@usf.edu

## ABSTRACT

Augmented reality (AR) allows visualizations to be situated where they are relevant, e.g., in a robot's operating environment or task space. Yet, headset-based AR suffers a scalability issue because every viewer must wear a headset. Projector-based spatial AR solves this problem by projecting augmentations onto the scene, e.g., recognized objects or navigation paths viewable to crowds. However, this solution mostly requires vertical flat surfaces that may not exist in open areas like warehouses, construction sites, and search and rescue scenes. Moreover, when humans are not co-located with the robot or situated at a distance, the projection may not be visible to humans. Thus, there is a need to create a flat, viewable surface for humans in such scenarios.

In this work, we designed a prototype of a fog screen device to create a flat, projectable display surface in the air. A robot with such a device mounted empowers it to communicate with humans in environments without flat surfaces or with only irregular surfaces in search and rescue sites. Specifically, we detail the design, covering underlying principles, material selection, 3D modeling, and evaluation. We will verify and iterate the design, explore the optimal mounting position, and conduct human evaluation in the future.

## CCS CONCEPTS

• Hardware → Displays and imagers; • Human-centered computing → Mixed / augmented reality; • Computer systems organization → Robotics.

## KEYWORDS

Fog screen, mid-air display, robot communication, projected AR, augmented reality (AR), human-robot interaction (HRI)

**ACM Reference Format:**
Uthman Tijani and Zhao Han. 2024. Prototyping Mid-Air Display for Anywhere Robot Communication With Projected Spatial AR . In *Proceedings of the 7th International Workshop on Virtual, Augmented, and Mixed-Reality for Human-Robot Interactions (VAM-HRI '24), March 11, 2024, Boulder, CO, USA.* 5 pages.

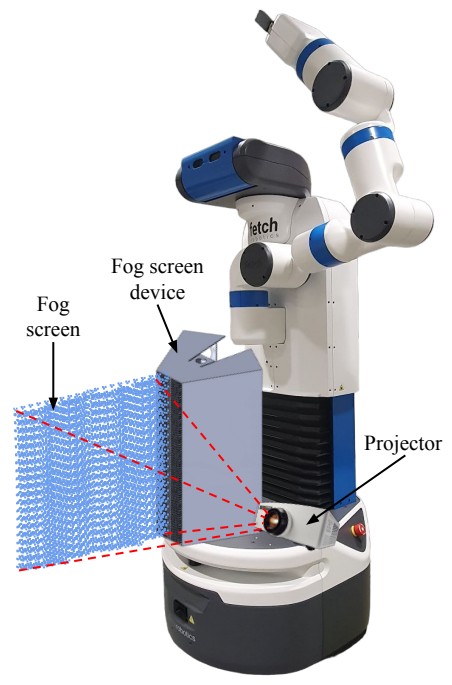

**Figure 1: Proposed conceptual setup of our system, a low-fidelity representation of our plan to place the fog screen device and the projector on the robot. Details of the fog screen device are in the next pages.**

## 1 INTRODUCTION

In HRI, various communication modalities have been explored [25], including speech and audio [1, 18], gaze [11], visual displays [23], and body language like gestures and postures [8]. Research efforts have started to leverage augmented reality (AR) to enhance robot communication of non-verbal cues [3–5, 7, 12, 17, 27], including projected AR works [2, 7] where a robot's navigation paths and manipulation intent were projected onto the ground and table surface.

Particularly, AR allows visual overlays situated in a robot's task environment to externalize a robot's internal states. Projector-based AR further improves AR as it offers scalability such that a crowd of people no longer need to wear AR headsets. However, it is crucial to consider different environments, e.g., settings that lack projectable surfaces or those not viewable by humans, instances where humans

are not co-located with or further away from the robot, and situations where physical obstructions may affect the visibility and effectiveness of AR visualizations. While these work [2–5, 7, 12, 17, 27] improve the understanding of a robot's intent and behaviors like manipulation and navigation using AR, we must solve the problem that some environments lack projectable surfaces.

To retain the benefits of projected AR, we must answer this question: "how can we leverage this technology for robot communication when there is no available or suitable (e.g., irregular) surface to project onto?" In response, we propose using a fog screen device to create a mid-air flat display medium for robots to communicate in environments lacking projectable surfaces. This is particularly beneficial for open environments like warehouses, construction sites where suitable surfaces may be scarce or search and rescue scenes with irregular surfaces.

Specifically, we present a prototype design of a flat fog screen device. Currently, we are building the first version of the device and verifying the optimal mounting position of the fog screen device, together with a projector, on a specific robot platform—the Fetch mobile manipulator. Our initial concept plan (Figure 1) is to place the fog screen device in the front right of the robot's base and place the projector in the front left of the base, therefore making it possible to project content onto the generated fog screen.

In the subsequent sections, we delve into the specific design considerations–how the fog screen device works–and evaluation plans to verify the proposed fog screen device with humans. Through this work, we strive to contribute practical insights and address communication challenges in dynamic environments.

## 2 RELATED WORK

### 2.1 Projected Augmented Reality (AR)

This section explores how projected AR enhances human-robot communication in various aspects, from conveying robot intentions to addressing challenges in understanding robot navigation in populated areas. For example, Chadalavada et al. [2] aimed to improve communication between humans and robots in collaborative workspaces. By projecting a robot's internal states onto the shared floor space, the paper investigates how AR can convey a robot's navigation intent. Han et al. [7] also tried to address the challenge of understanding the navigation intent of mobile robots in populated areas while also open-sourcing the hardware and software implementation for the projected AR technology. The emphasis was on directional projections, specifically arrow projections, as visual cues to convey the robot's intended navigation path. The goal was to provide a practical solution for visualizing robot navigation intent, contributing to improved participant perception, enhanced comfort, and behavioral changes in response to the projections. Our work also belongs to the projected AR family, but we focus on creating a fog screen for robot communication.

### 2.2 Mid-Air Display

The exploration of mid-air display technologies has led to diverse innovations addressing challenges in cost [10, 13], bulkiness and low resolution [13], fog flow deformation [14], tactile feedback [20, 21], and mounting position [6].

Rakkolainen et al. [16] explores the fog screen's potential application domains like entertainment, museums, and trade shows. They outlined future research directions to address challenges related to high costs, multi-modality interactions, and tracking using new sensors. Particularly related to robotics, Scheible et al. [22] introduced "Displaydrone", a system that has a drone, a Pico projector, and a smartphone to create a flying interactive display for projecting content onto walls and objects in physical space, such as building facades and rock surfaces. This work highlights challenges in mobile interactive displays, such as short battery life, legal issues, projection viewing distance, and stable hovering. Finally, Otao and Koga [14] addressed the difficulty of detecting the deformation of fog flow caused by a user's action in real-time by proposing a feed-forward approach to create pseudo-synchronized image contents along with the deformed fog.

### 2.3 Feedback for Mid-Air Display Interactions

To make user interaction with fog screen display responsive, Sand et al. [20, 21] investigated the effect of wearable vibrotactile feedback devices for mid-air gestural interaction with large fog screens on user preference [21]. Two finger-based gestures, tapping and dwell-based (i.e., the user's finger stays longer in the mid-air display) were compared. A user study showed that tapping without haptic feedback was the most preferred gesture type, while haptic feedback was preferred for the dwell-based gesture. Similarly, Dancu et al. [6] investigated optimal wearable mid-air display placement during map navigation, comparing wrist and chest mounting approaches. Results from the user study showed that wrist-mounting is considered safer and preferable for map navigation.

### 2.4 Reconfigurable Mid-Air 3D Display

Researchers also attempted to display volumetric 3D content by creating 3D fog displays. Lam et al. [10] used columns of upward-flowing laminar fog which is reconfigurable (fog particles can be re-positioned), allowing for true 3D perception without AR headsets, as projected images scatter at different depths. Lam et al. [9] later improved their work [10] by achieving a higher resolution, enabling users to touch and manipulate virtual objects with wide viewing angles directly. Tokuda et al. [24] also introduced combining shape-changing interfaces and mid-air fog displays, which can adapt their shape to support 2D and 3D visual content.

### 2.5 Compact Cost-Effective Mid-Air Display

While most works focused on user interaction [14, 20, 21], shape-changing, reconfigurable fog screens and volumetric 3D content display [9, 10, 24], Norasikin et al. [13] addressed the issue of high cost, bulkiness, and inability to provide higher resolution of mid-air displays. SonicSpray [13] addressed these challenges by offering a compact, low-cost solution that can precisely control laminar fog (suspension of tiny solid or liquid particles in gas) in mid-air. As some of the works [15, 20, 21] focused on tactile feedback for fog screen displays, our focus is mainly on creating compact fog screen displays for robot communication. So far, fog screen creation has not been explored for communication in the HRI community.

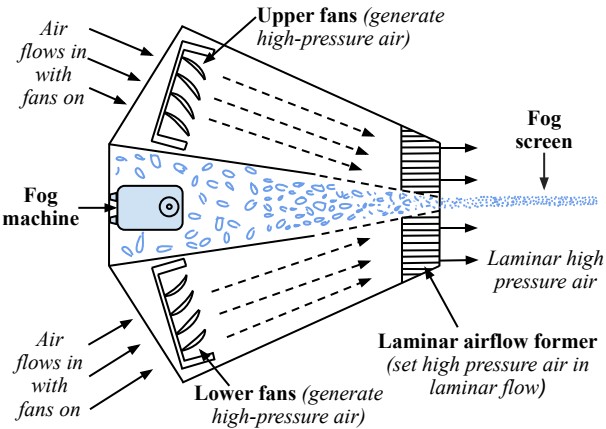

Figure 2: Top view of our fog screen device showing its internal mechanism. The blue particles within the device generated by the fog machine fill the middle container and exit the right opening. The laminar airflow former (Figure 3) regulates the dispersed fog into a laminar flow to form a screen.

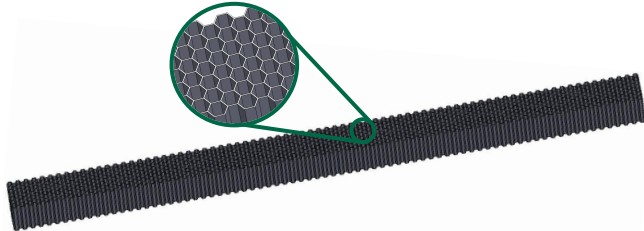

Figure 3: 3D CAD model of laminar airflow former, passage for high-pressure air from the fans to make the fog exit in a particular direction to form a flat screen. Taking a closer look at the airflow former design, it is similar to gluing a series of plastic straws together by the side or like a honeycomb.

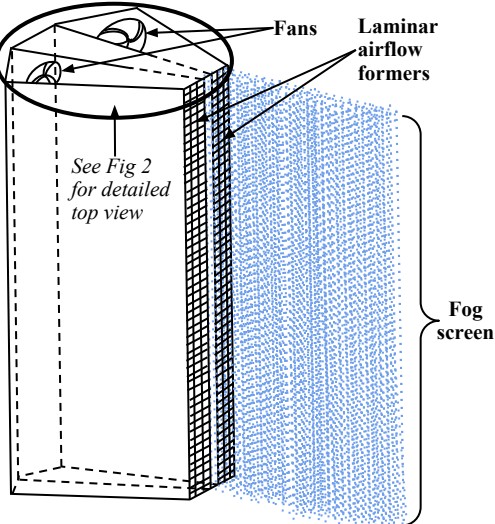

Figure 4: Full side view of our fog screen device with fans. This is how it will be positioned on the robot, allowing a wide viewing and projection angle.

## 3 DESIGN

Our goal is to design a compact, portable fog screen device that can be mounted conveniently on a Fetch robot to be projected onto by a projector, as seen in Figure 1.

As seen in Figure 2, the device begins by making water into tiny droplets to generate an uncontrolled mist or fog, then uses a set of fans to generate high-pressure air, and finally uses the laminar airflow formers to transform the fog into a stable and flat fog screen.

First, we used an off-the-shelf fog machine to generate fog with a special liquid called "fog juice" [26]. It consists of water, glycerin, and propylene glycol. When this liquid is heated, it turns into mist or fog. A built-in pump pushes this mist through a small opening on the right. When it meets the cooler air outside, it turns into opaque fog that can be seen. Although the fog can be produced by an ultrasonic atomizer placed in a container of water, we realized that using the fog machine is cleaner (no concerns for water leakage), flexible (changeable device orientation), and produces denser fog using fog juice (mixtures of water and glycerin) [26].

The middle fog container will host the fog machine. In the container, the generated fog will accumulate first before coming out of the thin opening on the right. Because the generated fog at this stage is dispersed as it comes out, it is unsuitable to project onto as it does not make it a flat screen. To solve this, it must be controlled to follow a laminar flow resembling a thin layer of surface in mid-air for projection.

The set of fans at both sides (Figure 2 top and bottom) and the airflow formers (Figure 2 right and Figure 3) are designed to achieve this. The fans generate high-pressure air by sucking in air through the vents at the left end of the device and traveling right to the airflow formers, where the airflow is made laminar. These laminar high-pressure air on both sides serve as a barrier for the fog, forcing it to remain within it, creating a flat, thin layer of fog in mid-air suitable for projectors to project onto.

Figure 4 shows a full side view of the device, giving a closer look at the smaller view in Figure 2.

### 3.1 Materials

We plan to test our prototype by first 3D-printing all the parts we have designed. Besides the laminar airflow former shown in Figure, we have also designed a 3D CAD model of the case using SOLID-WORKS, which is shown in Figure 5. The case houses and protects the internal components like the fans and external components like the airflow formers. This initial case design was inspired by Hover-lay II [28]. Additionally, we plan to buy the fans, the fog machine, and a portable battery to power the fans and the fog machine.

### 3.2 Placement on Robot

The housing design structure and the fog machine allow the device to be placed vertically as shown in Figure 1. We plan to place the fog screen device on a Fetch robot's base, which has M5 holes with

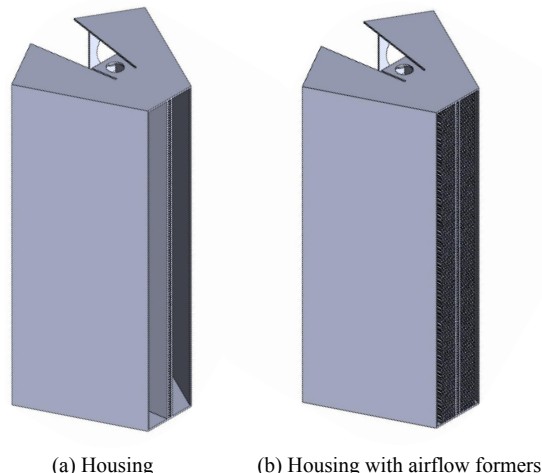

|             |                                |
|:-----------:|:------------------------------:|
| (a) Housing | (b) Housing with airflow formers |

**Figure 5: 3D CAD model of the case of the proposed fog screen device, about 640mm wide. (a) The housing is more like the device framing. It has slots for the airflow formers and the fog container. (b) This is how the device looks after all designed parts have been assembled (housing, airflow formers, and fog container).**

a grid spacing of 50$mm$ [19]. The projector will also be placed on the base near the fog screen device.

## 4  EVALUATION PLAN

First, we will build the prototype with all the parts 3D printed or bought and verify our theory to see if the device can produce a desirable flat fog screen. This may take several iterations. We will examine whether the airflow former can prevent the fog from dispersing and keep the fog flow laminar, i.e., keep the airflow former parallel to the fog emission points. A slight error in the angle will disperse the fog from a flat screen shape. We will also examine how wide the screen can be and adjust accordingly.

Our next step is to verify the responsiveness of the fog screen generation. We will measure the time the fog machine and the fog screen device take to begin generating fog.

Third, we plan to test reliability for consistently generating usable fog for projection use. Here, we determine the performance or efficiency change over extended periods of use. The changes in the fog density and how long the fog machine can keep producing dense and wide, i.e., usable, fog will be closely monitored.

Fourth, we plan to test the robustness of the generated fog in different wind conditions to make it suitable for outdoor use. We expect that the high-pressure wind generated by the airflow former helps achieve robustness.

Formal human evaluation would be the final phase, getting user perception and feedback for potential design guidelines and improvements. In the far future, we also plan to explore ways to provide privacy between the user and the robot by either stopping the projector or dispersing the fog in the presence of a third party.

## 5  CONCLUSION

We proposed a prototype design of a fog screen device to make a mid-air display surface to address the difficulty of robots leveraging scalable projected AR to communicate in environments lacking projectable surfaces. We plan to build and iterate the device and mount it onto a Fetch robot. In the future, we will further test its functionality, reliability, and user perception.

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
