# OpenReview forum: "Prototyping Mid-Air Display for Anywhere Robot Communication With Projected Spatial AR"
_humanrobotinteraction.org/HRI/2024/Workshop/VAM-HRI — HRI 2024 Workshop VAMHRI Withdrawn Submission_

### Official Review · Reviewer_Ahdr · 2024-02-01
**Accept**

**Rating:** 7
**Confidence:** 5

**Review:**

This paper introduces a fog screen device prototype to enable mid-air displays for robot communication in environments lacking suitable projection surfaces. This approach addresses the limitations of projector-based spatial AR by providing a portable, visible surface for augmented reality (AR) projections, which is especially beneficial in open spaces or areas with irregular surfaces.

### Strengths:
- Innovative solution for enhancing human-robot interaction through spatial AR in challenging environments.
- Detailed design and prototyping methodology, including material selection and 3D modeling.

### Weaknesses:
- The paper focuses on the prototype's design and theoretical application, with future work needed to evaluate its practical effectiveness.
- Relies on specific hardware (fog screen device and projector), potentially limiting more comprehensive application or adaptability.

In summary, I think this paper is a great fit for VAM-HRI, and I recommend acceptance.

---

### Official Review · Reviewer_DCSz · 2024-02-02
**Accept**

**Rating:** 7
**Confidence:** 5

**Review:**

The paper explores the development of a fog screen device as a prototype for mid-air display, aiming to enable robot communication in environments lacking projectable surfaces. The motivation comes from the limitations of traditional augmented reality (AR) systems, particularly in open areas like warehouses, construction sites, and search and rescue scenes. The proposed fog screen device, mounted on a robot, creates a flat, projectable display surface, which when used in combination with a projector allows for improved human-robot interaction. The paper explains the design aspects of the fog screen device and planned evaluation. The proposed implementation is planned to be used with the Fetch mobile manipulator in environments like warehouses, construction sites, or search and rescue scenes. There is also mention of human-in-the-loop experiments to assess user perception and feedback.


Strengths:

-Innovative Concept: The paper introduces a method to overcome the limitations of conventional AR systems by proposing a fog screen device for mid-air display.

-Design Considerations: The paper explains the fog screen device's design, covering underlying principles, material selection, modeling, and evaluation plans.

-Application potential: The proposed fog screen device offers a potentially scalable solution for robot communication across mobile platforms, especially in scenarios where traditional AR systems face challenges. The scalability of this technology could address some of the limitations of AR technology for HRI applications.


Weaknesses:


-The statement about exploring ways to provide privacy by stopping the projector or dispersing fog in the presence of a third party is unclear, especially considering the initial goal of addressing scalability issues with AR headsets.


-The limitations of the fog machine are unclear. Potential issues could be water capacity, maintenance, device usage duration, fog screen coverage, and susceptibility to environmental factors like wind and lighting conditions.


-Due to a lack of experimental implementations, it is unclear about the outcome of practical usability and human-in-loop applications.

In summary, I think this paper is a good fit for VAM-HRI, and I recommend acceptance.

---

### Decision · Program_Chairs · 2024-02-06

Accept (Oral)